# Grazing Cow Behavior’s Association with Mild and Moderate Lameness

**DOI:** 10.3390/ani10040661

**Published:** 2020-04-11

**Authors:** Niall W. O’Leary, Daire. T. Byrne, Pauline Garcia, Jessica Werner, Morgan Cabedoche, Laurence Shalloo

**Affiliations:** 1Land Management and Systems, Faculty of Agribusiness and Commerce, Lincoln University, Lincoln, 7647 Christchurch, New Zealand; 2Animal and Grassland Research and Innovation Centre, Teagasc, Moorepark, Fermoy, P61 C997 Cork, Ireland; thomas.byrne@teagasc.ie (D.T.B.); laurence.shalloo@teagasc.ie (L.S.); 3Seenovate, MIBI Building 672, Rue du Mas de Verchant, 34000 Montpellier, France; pauline.garcia34@yahoo.fr; 4Animal Nutrition and Rangeland Management in the Tropics and Subtropics, University of Hohenheim, 70599 Stuttgart, Germany; jessica.werner@uni-hohenheim.de; 5IUT de Vannes, 8 Rue Michel de Montaigne, 56000 Vannes, France; MORGAN2509@live.fr

**Keywords:** accelerometer, lameness, pasture, behavior, dairy cow, mobility, locomotion

## Abstract

**Simple Summary:**

Diseases of cow hooves usually affect how a cow walks (gait), and cows with sufficiently abnormal gait are classified as lame. In addition to altered gait, there are also reports of differences in behavior, such as lying time. To aid heat detection, cow behavior is regularly measured on farms using cow-attached accelerometers. Using these and similar measures of cow behavior to detect lameness has been investigated with mixed results. The strongest reported lameness–behavior associations have been with lying time and activity measures. Most of these and other associations have been reported previously with zero-grazing cows. Here, we looked to see if 14 different behavior measures were associated with mild and moderate lameness in grazing animals. Four trials were performed with two breeds of cows across two farms with data from 63 cows aggregated together in total. Measures of activity and standing/laying events were weakly associated with lameness. The usefulness of these measures for lameness detection with grazing animals may thus be limited.

**Abstract:**

Accelerometer-based mobility scoring has focused on cow behaviors such as lying and walking. Accuracy levels as high as 91% have been previously reported. However, there has been limited replication of results. Here, measures previously identified as indicative of mobility, such as lying bouts and walking time, were examined. On a research farm and a commercial farm, 63 grazing cows’ behavior was monitored in four trials (16, 16, 16, and 15 cows) using leg-worn accelerometers. Seventeen good mobility (score 0), 23 imperfect mobility (score 1), and 22 mildly impaired mobility (score 2) cows were monitored. Only modest associations with activity, standing, and lying events were found. Thus, behavior monitoring appears to be insufficient to discern mildly and moderately impaired mobility of grazing cows.

## 1. Introduction

Abnormal walking is a symptom of a wide range of painful pathologies affecting limbs of cows [1,2,3]. Lameness, impaired mobility, impaired locomotion, or abnormal gait are ways of referring to abnormal walking. O’Connor et al. [3] defined suboptimal mobility as ‘any abnormality to a cow’s gait that causes a deviation from the optimal walking pattern of a cow’. Van Nuffel et al., [4] defined lameness as ‘the clinical manifestation of painful disorders, mainly related to the locomotor system, resulting in impaired movement or deviation from normal gait or posture’. The threshold at which a cow is defined as lame can vary from scale to scale or person to person and so is somewhat arbitrary [5]. Horseman et al., [6] documented the deliberate moving away from the term lameness to mobility scoring in the UK in order to make the topic more palatable to farmers. However, this might be misleading and farmers could be taking ‘mobility’ less seriously than ‘lameness’ [6]. We refer to mobility when discussing the act of scoring cows, but otherwise refer to lameness.

Lameness is endemic in dairy herds and is a major welfare issue for grazing cows [3,7]. However, lameness is not routinely measured on many farms [1]. Farmer estimates of their herd’s lameness are usually only a third, or even just a quarter of the true prevalence. The prevalence of severe lameness aligns with farmers’ estimates of total lameness [2,7] indicating many farmers are not conscious of mild and moderate lameness in their herds. This information gap relating to mild and moderate lameness could be a major barrier to lameness management. In particular, mild cases could be treated earlier, preventing increasing severity and speeding recovery. Regular whole herd assessments of mobility, either manually or automated, would thus be valuable.

However, manual observation is time-consuming, subjective, and requires training [8]. Manual mobility scoring entails assessing gait asymmetry, stride length, reluctance to bear weight, back arch, and walking speed [5,9]. Various automated approaches have been trialed which measure a small subset of these or novel indicators of mobility relative to manual observers. These automated approaches to collecting indicators of mobility include cows walking over pressure-measuring plates or mats, computer vision, and accelerometers [10,11]. However, commercially available systems for farmers remain rare [12]. Those that are available have not been independently validated. Accelerometers are increasingly used on dairy farms for heat detection [13]. Heat detection is achieved by measuring changes in behavior (such as activity) relative to herd mates and relative to individual cows’ past behavior. Using similar behavior measures to detect lameness has been a prominent area of research [14,15]. Beer et al., [16] reported 91% accuracy using both behavior (standing bouts) and a gait measure (walking speed) for zero-grazing (housed) cows. Thorup et al. [14] reported that walking duration and sensor movement was associated with lameness for zero-grazing cows, but did not report lameness classification accuracy. 

The specific behaviors examined, grazing or zero-grazing, lameness severity, and cow breed have varied between studies. Most behavior-focused studies have included severely lame cows [15,16] and have usually studied zero-grazing cows with the study of grazing cows being the exception [15,17,18,19,20]. Similar but weaker associations have been reported for grazing cows as for zero grazing cows. Kamphuis et al [15] reported that step frequency was associated with lameness for grazing cows. Byabazaire et al., [17] reported lameness classification accuracy of up to 87% for grazing cows. They did this by measuring step frequency in addition to lying time and changes from one behavior to another over an extended period. Grazing and zero-grazing cow behavior may not be directly comparable. For lying time in particular, there is evidence that lying time is affected by precipitation with grazing cows lying less on wet pasture [20].

Blackie et al [21] reported differences in lying time between zero-grazing lame and nonlame Holsteins. Subsequently, Blackie and Maclaurin [22] in a similar study did not find similar associations with Jersey cows. Behavior–lameness associations could thus fail to generalize depending on the severity of lameness, grazing/zero-grazing, and breed. Furthermore, some studies have found weak, no, or conflicting associations, in particular for lying time [14,23,24,25]. The feasibility of behavior-based approaches for detecting lameness thus remains unproven. 

This paper monitors 14 behaviors of grazing Holstein and Jersey cows, and we then assess if these are indicative of mildly and moderately impaired mobility. This focus on mild and moderate lameness in grazing cows is novel. A further novelty is the inclusion of both Holstein Friesians and Jerseys. Finally, the trials were carried out in commercially relevant conditions. In commercial applications, controlling for many behavior confounding variables will be impractical. The goal is thus not to discern if a behavior is associated with lameness in controlled conditions. Instead, we investigate, are these associations reliably discernible in commercially relevant contexts? If yes, then they are likely to be useful for automated lameness detection of mild and moderate lameness for grazing animals.

## 2. Materials and Methods 

The Moorepark Animal Welfare Body was consulted regarding the ethical implications of the present study. As the study was observational with no invasive procedures, formal ethical approval was not required. Four groups of cows were mobility scored to select subsets of animals stratified by mobility score to monitor behavior from two farms. All cows were lactating and grazing with minimal supplementation (0–2 kg/day). Cows were spring-calving and the trials occurred between June to August. Trials were midlactation, after breeding, and thus unlikely to be affected by estrus. The days where behavior was examined had almost no precipitation (Trial 1.A 0.7 mm, 1B. 0.1 mm, other trials 0 mm). Cows walked to and from the paddock twice daily for milking. Paddock to parlor distance varied from day to day but did not exceed 1.5 kilometers. 

A 0 to 3 ‘mobility’ scale was used to score cows [26]. In this system, zero is good mobility, 1 is imperfect mobility, 2 is impaired mobility, and 3 indicates severely impaired mobility [26]. We interpret the mobility scores as being equivalent to the following: 0 as nonlame, 1 as mild lameness, 2 as moderate lameness, and 3 as severe lameness. One experienced technician who teaches mobility scoring, scored cows as they exited the milking parlor. As mildly and moderately lame cows were identified, they were drafted for inclusion into the study until six or seven of each had been selected. Nonlame cows were selected so that they had similar milking order to lame cows. Studied cows remained in their normal group and each trial consisted of cows managed together in groups of between 40 and 200 cows. Trial sample sizes were limited by available pedometers (initially 21). To ameliorate this, four trials were conducted reusing the pedometers (Table 1). Three trials were conducted at Dairygold research farm, Cork, Ireland. Of the three Dairygold trials, one was with purebred Jersey and two were with Holstein Friesian cows. The fourth trial was at a nearby commercial farm with Holstein Friesian cows. For trials 1, 3, and 4, cows were mobility scored at the beginning (A) and the end (B) of these trials (Table 1). The time between scoring events ‘A’ and ‘B’ varied with operational requirements from 4 days to 14 days. In trial 2, cows were only scored once due to locomotion scorer unavailability. 

The pedometers used to record behavior in this study were the RumiWatch 10 Hz pedometer (Itin + Hoch GmbH, Liestal, Switzerland). RumiWatch accuracy for lying, standing, and walking activity has been reported as sufficient for research purposes [27,28], and the RumiWatch has previously been used successfully in one study to detect lameness in zero-grazing cows [16]. A pedometer was attached to a rear leg at the metatarsus position for each observed cow. Daily summaries of 14 behaviors (Table 2) were generated using the RumiWatch Converter 7.3.36, algorithm V00_56 [29]. The mean and standard deviation of the 24-hour summaries for each of the 14 measures for each of the seven scoring events is presented (Table 3). 

Operator errors and technical faults resulted in smaller than expected sample sizes. Some pedometers failed to operate at all. One appeared to operate but failed to record data. Three recorded erroneous data with no or unfeasibly low walking time and these were excluded from the analysis. The first three trials had 16 and the final trial had 15 useable data sets (Table 1). Samples were of mixed parity. In trial 1, five cows were first lactation and 11 were multiparous. In trial 2, there was one cow with a parity of 1 and 15 cows were multiparous. In trial 3, all the cows were multiparous. In trial 4, three cows had a parity of 1 and 12 cows were multiparous. 

Power analysis was performed to inform the interpretation of the findings using the R package pwr [30]. Spearman’s Rho correlation was calculated for behaviors and mobility score in individual trials. Generalized linear models were used to assess associations in aggregated data. Mobility score was the dependent variable while the “behavior” (e.g., lying bouts) and trial were the fixed effects. The data and the r code used in this study are available online via github [31]. 

**Table 2 animals-10-00661-t002:** Variables analyzed and definitions.

Variable	Definition	Reference
Activity	Activity index (without dimension), proportional to the variability of the three acceleration axes.	[28]
Laydown	Lie down instances (pedometer angle changes from a vertical angle to a horizontal angle for at least 50 s) within the summary time frame.	[27,28]
Laying Counter	The number of periods with the pedometer in a horizontal position >50 s. Interruption of this pedometer position for less than 50 s is identified and calculated as one stand-up and one lying-down event but not as a separate standing bout.	
Laying Index	Activity index while lying.	
Lay Time	Sum of the duration of all lying bouts within a given recording period.	[27,28]
Limb Events	Movements of the legs or < 3 strides, no./time frame.	[32]
Standing Counter	The number of periods during which the cow is in an upright position but not walking. A temporary change from a vertical angle for less than 50 s is neither rated as lying-down and standing-up events nor as an additional lying bout.	
Standing Index	Activity index while standing.	
Stand Time	Standing time slice (time sum in minutes) within the summary time frame.	[27,28]
Stand Up	Get up instances count within the summary time frame.	[27,28]
Strides	One forward or backward movement of the limb within a walking bout.	[28]
Walking Counter	The number of periods characterized by at least three consecutive strides in the same direction (forward or backward). The period betweentwo strides must not exceed 4 s. Walking bouts are rated as separate if the time between two strides exceeds 10 s.	
Walking Index	Activity index while walking.	[29]
Walk Time	Walking time slice (time sum in minutes) within the summary time frame.	[27,28]

**Table 3 animals-10-00661-t003:** 24-hour summary statistics for cows in each trial—mean and standard deviation (Sd).

	1.A Jerseys		1.B Jerseys		2. Holstein Friesian 2017		3.A Holstein Friesian 2018a		3.B Holstein Friesian 2018b		4.A Farm		4.B Farm	
Variable	Mean	Sd	Mean	Sd	Mean	Sd	Mean	Sd	Mean	Sd	Mean	Sd	Mean	Sd
Activity	244	29	156	21	197	29	147	18	150	20	111	11	135	7
Laydown (n/day)	7.3	1.5	7.4	1.9	9	1.7	8.9	1.3	8.6	2	8.1	1.9	8.7	2.3
Laying Counter	7.1	1.4	7.4	1.8	8.9	1.5	8.9	1.2	8.4	1.8	8.1	2.1	8.6	2.3
Laying Index	5.9	1.4	7.2	2.6	10.3	2.8	7.9	2.3	9.5	3.2	8.6	1.4	8.9	2.5
Lay Time (min/day)	616	89	622	131	564	91	619	57	542	66	738	69	683	71
Limb Events (n/day)	2910	591	2066	520	1982	345	2011	287	2393	320	1716	334	1895	277
Standing Counter	307	51	245	39	209	39	222	28	235	45	217	20	201	23
Standing Index	250	64	138	21	119	21	121	21	127	18	104	16	116	15
Stand Time (min/day)	690	80	714	125	748	89	722	57	799	65	621	70	670	72
Stand Up (n/day)	7.3	1.6	7.3	1.7	9.1	1.7	9	1.5	8.8	2	8.3	1.8	8.5	2.1
Strides (n/day)	4259	707	3152	486	4402	333	2943	249	2873	369	2296	333	2662	223
Walking Counter	304	52	239	39	201	39	215	28	229	46	213	21	194	23
Walking Index	1317	132	1178	113	1479	183	1212	108	1106	84	1092	101	1263	137
Walk Time (min/day)	134	18	104	16	129	12	99	10	100	15	82	11	88	9

## 3. Results

Table 3 summarizes the mean and standard deviation of each behavior variable for each mobility scoring event. It shows that mean behavior varied between and within trials. Table 4 reports the statistical sensitivity or minimum detectable effect sizes for each data set. Trial 4 with 15 cows was sufficiently powered to detect correlation coefficients of 0.53 and higher (alpha = 0.1) [33]. None of the 14 behavior measures were detected as consistently associated with mobility score across individual trials (Table 4). 

In addition to analyzing individual trial data, the individual trial data was also aggregated/pooled as follows. Trials 1A, 2, 3A, and 4a were combined into one dataset, henceforth known as aggregated dataset A. Trials 1B, 2, 3B, and 4B were combined into another dataset labeled as aggregated dataset B. With 62/63 observations, these datasets were sufficiently powered (f2 = 0.16) to detect what Ford [33] described as ‘medium’ effect sizes with an alpha of 0.1 in a general linear model with two independent variables. This would be a comparable effect size to a Pearson’s correlation of ~0.31 [33]. 

For aggregated dataset A only, Laying Counter and Stand Up were significantly associated with mobility. Laying counter and Stand Up counter are almost identical measures with a Pearson’s correlation of 0.94 between them in aggregated dataset A. Each standard deviation more laying counter/standup events predicted 0.29 standard deviation lower mobility score. The mobility score standard deviation in this sample was 0.8, so a standard deviation more laying counter predicted 0.23 (0.8*0.29) lower mobility score.

Figure 1 shows the modest differences in laying counter by mobility score from aggregated dataset A. For aggregated dataset B, neither Laying Counter nor Stand Up were associated with mobility score. In aggregated dataset B, Activity and Standing Index had a standardized coefficient to mobility score of −0.38 and −0.28, respectively, and were statistically significant (*p* < 0.05). Both are measures of activity, the former through the whole day, the latter while the cow was standing. Similar standardized coefficients were found in aggregated dataset A. However, these associations were not statistically significant. In this study, parity was not associated with mobility score and was not correlated with any of the 14 behavior measures in this study. 

In summary, none of the assessed behaviors were consistently associated with mobility-score across the four individual trials. Stand up/laying counter had a significant association with mobility in aggregated dataset A. This is consistent with another Irish study with grazing cows where changes in behavior were indicative of lameness [17]. This association was only statistically significant in one individual trial (1.A). Contradictory nonsignificant positive correlation coefficients were found in individual trials 1.B and 2. Activity and activity while standing (Standing Index) were the strongest indicators of mobility score in this study. They both had relatively high standardized coefficients with mobility in both aggregated datasets A and B. However, these were not statistically significant in the analysis of aggregated dataset A. Even these are thus unreliable or intermittent indicators of mobility score. 

## 4. Discussion

### 4.1. Main Findings

To date, the variables most used to assess lameness using accelerometers have been differences in behavior [34]. Contributing to mixed literature, the assessed behavior measures were not reliably associated with mobility scores. Based on the present and other results [14,20,23,24,25,34], measuring lying time appears to be of limited value for lameness detection. Activity and Standing Index were indicative of mobility scores in both aggregated datasets A and B, but only statistically significant in B. These modest associations indicate these behavior variables could only play a small part in a mild and moderate lameness detection system for grazing cows. 

### 4.2. Study Limitations

In this study, the time between scoring and measurement (Table 1) could have obscured associations. This study also only used one, albeit highly trained, mobility scorer. Other studies have used the mean of multiple scorers’ assessments, made efforts to improve alignment between scorers [35,36], performed hoof inspections, and recorded video of cow locomotion [35]. Video based scoring has the benefit of replay and playback speed adjustment while also being noninvasive. Both video and hoof inspections potentially allow for better identification of the affected limb, which may be important. Future research should consider using multiple scorers, hoof inspections, and video recordings. 

The severity of lameness and how easily it can be detected using behavior are likely to be associated. Some studies [18,22,37] have investigated a greater range of lameness severity than the present study. The present findings indicate that mild and moderate lameness is difficult to detect, whereas severe lameness detection might be more easily achieved. Severe lameness detection would be of value for animal welfare auditing. However, as severely lame cows are usually readily identified by untrained staff, severe lameness detection may be of less value for lameness management than mild and moderate lameness detection. 

Future research might manage different breeds together to control for management differences. No consistent patterns here could be discerned with either Holstein Friesian or Jersey cows in this study. Thus, inferences cannot be made about the differences between them in terms of behavior and lameness. 

The trial sample sizes in this study were not unusual. Past studies have reported sample sizes of 9 [38], 10 [39], and 59 [21]. Sixty-three cows across four trials appears to have been underpowered to replicate previously reported associations in the current context. In particular, modest correlations of less than 0.3 were unlikely to be confirmed. This and the focus on mild and moderate lameness could explain the modest findings in this study. Nevertheless, the findings from an application-relevant context provide a strong indication that these behaviors are unlikely to be very useful for mild and moderate lameness detection for grazing cows. That more associations might have been statistically significant with a larger sample is probable. However, the small effect size and variability implied by the current findings indicate they would be of limited value for mild and moderate lameness detection. Other studies have included severe lameness, observed cows over longer periods, and established baseline levels of behavior against which relatively more promising lameness detection performance has been achieved [17]. However, the highest accuracy using similar technology achieved 91% lameness detection accuracy by only assessing three days of behavior [16]. Extended observation and observation relative to herd mates may facilitate finding stronger associations [34] and should be considered in the design of future studies.

### 4.3. Implications of Findings

We set out to test if previous findings generalized to mild and moderately lame grazing animals. However, consistent patterns of association were not found. Two past studies of grazing animals have found associations between lying time and lameness [18,20] which was not found in this study. 

Between and within-trial behavior varied to a great extent (Table 3). Grazing cow management varies more than zero-grazing, especiallyfor distances walked to and from milking. This variation will confound behavior-based approaches. In grazing systems, significant challenges remain in creating effective automated lameness detection systems. To be useful for detecting lameness, a behavior measure should ideally be consistently associated with mild or moderate lameness in normal grazing conditions. This study indicates the studied variables fall short of this criterion. Future studies may wish to achieve greater statistical power so that smaller associations equivalent to a correlation of 0.2 at an alpha of 0.05 could be discerned. This would entail a sample size of approximately 150 cows. Such small effect sizes are, however, unlikely to be sufficient for accurate lameness detection. Novel measures of cow gait, such as stride duration [16], may be more indicative of lameness than measures of behavior [34], especially if gait measures are less influenced by management and as measures of gait are closer proxies of lameness than behavior. After all, it is upon gait that manual mobility scoring is based [9], not measures of behavior. Efforts to create effective ways to measure gait should thus be a key focus for accelerometer-based lameness detection efforts.

## 5. Conclusions 

This study set out to assess the generalizability of behavior—lameness associations reported in previous studies. Specifically, we attempted to determine if grazing cow behavior is associated with mild and moderate lameness. Measures assessed included lying time, walking time, behavior changes, and bout durations of behavior. Only measures of Activity and activity while standing (Standing Index), Laying down counter, and Stand Up counter were associated with mobility score, albeit inconsistently. The limited replication of previously reported associations was unexpected and raises questions about the feasibility of using automated behavior measurement as the foundation of automated mild or moderate lameness detection for grazing cows. Behavior is likely to be useful as a complement to other, as yet undeveloped, predictors of lameness such as measures of gait.

## Figures and Tables

**Figure 1 animals-10-00661-f001:**
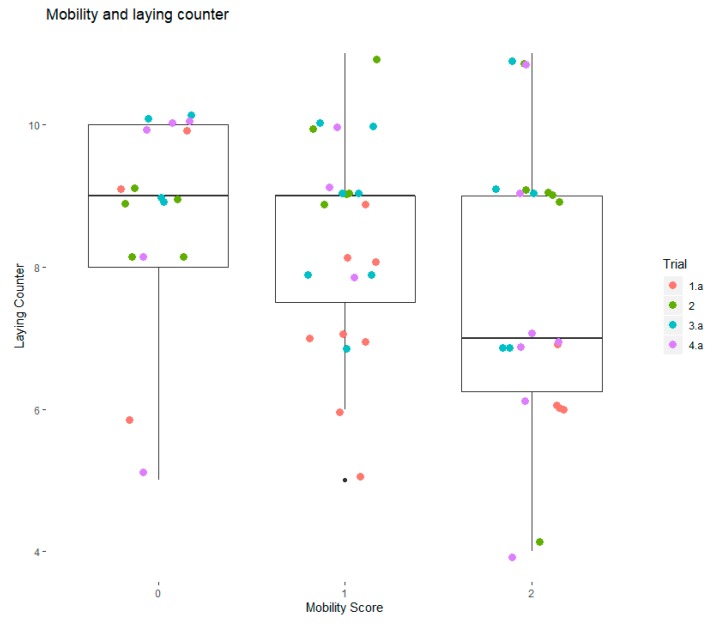
Laying counter (n/d) and mobility score for the aggregated dataset A. Laying counter is the number of times a day in which the pedometer was in a horizontal position for more than 50 seconds.

**Table 1 animals-10-00661-t001:** Trial descriptions. N is the number of datasets from each locomotion scoring event. M relates to the mobility score and the number of available datasets from cows of each score.

Trial, Location, and Date	N	M0	M1	M2	M3	Breed	Behavior Analysis Period Relative to When Cows Were Mobility Scored.
1. A* Dairygold (1 June 2017)	15	3	8	4	0	Jersey	Scored morning of 1 June 2017 and pedometers attached in the evening. Next day 24-hour behavior used (2 June 2017).
1.B* Dairygold (15 June 2017)	16	2	8	5	1	Jersey	Pedometers were removed a day before scoring (14 June 2017). 24-hour summary analyzed from two days before scoring (13 June 2017).
2 Dairygold (16 June 2017)	16	5	5	6	0	Holstein Friesian	Scored and pedometers attached morning of 15th. The next day’s behavior analyzed (16 June 2017).
3.A* Dairygold (8 August 2018)	16	4	7	5	0	Holstein Friesian	Scored in the morning and pedometers attached in the evening (8 August 2018). 24-hour summary from two days later analyzed (10 August 2018).
3.B* Dairygold (13 August 2018)	16	4	6	6	0	Holstein Friesian	24 hour summary from the same day as scoring analyzed (13 August 2018).
4.A* Commercial farm (16 August 2018)	15	5	3	7	0	Holstein Friesian	Scoring and pedometers attached in the morning (15 August 2018). The next day (24-hour summary) analyzed (16 August 2018).
4.B* Commercial farm (20 August 2018)	15	6	3	6	0	Holstein Friesian	Scored in the morning. 24-hour summary the day of scoring analyzed (20 August 2018).
1.A*, 2, 3.A* and 4A* aggregated	62	17	23	22	0	Jersey and Holstein Friesian	
1.B*, 2, 3.B* and 4B* aggregated	63	17	22	23	1	Jersey and Holstein Friesian	

* A and B refer to the first and second locomotion scoring events in a trial.

**Table 4 animals-10-00661-t004:** Spearman’s Rho correlation and standardized coefficients of behavior measures to mobility score.

Variable	1.A Jerseys	1.B Jerseys	2 Holstein Friesian 2017	3.A Holstein Friesian 2018a	3.B Holstein Friesian 2018b	4.A Farm	4.B Farm	Aggregated Dataset A Standardised Coefficients	Aggregated Dataset B Standardised Coefficients
n	15	16	16	16	16	15	15	62	63
Sensitivity ^	0.53	0.51	0.51	0.51	0.51	0.53	0.53	~0.31	~0.31
Activity	−0.06	−0.32	−0.62 *	0.03	−0.1	0.28	0.41	−0.28	−0.38 *
Laydown	−0.61 *	0.06	0.22	−0.29	−0.03	−0.34	−0.03	−0.28 †	0.07
Laying Counter	−0.47 †	0.06	0.21	−0.29	−0.11	−0.35	−0.04	−0.30 *	0.05
Laying Index	−0.03	−0.06	−0.26	0.33	0.21	−0.28	0.28	−0.05	0.10
Lay Time	−0.17	0.07	−0.01	−0.16	−0.02	−0.59 *	−0.09	−0.30 †	−0.06
Limb Events	0.04	0.07	0.04	−0.02	0.39	0.75 **	−0.03	0.27	0.13
Standing Counter	−0.01	−0.21	−0.05	0.03	−0.2	−0.15	0.08	−0.08	−0.20
Standing Index	−0.34	−0.19	−0.48 †	−0.06	−0.01	0.65 *	−0.17	−0.29	−0.28 *
Stand Time	0.08	−0.1	0.05	0.2	0.1	0.75 **	0.14	0.29	0.08
Stand Up	−0.47 †	0.03	−0.17	−0.29	−0.03	−0.32	−0.03	−0.29 *	0.04
Strides	0.36	−0.09	−0.22	0.01	−0.31	−0.43	0.23	0.00	−0.34
Walking Counter	−0.01	−0.23	−0.06	0.06	−0.18	−0.08	0.14	−0.07	−0.20
Walking Index	0.31	−0.5*	−0.47 †	0.31	0.06	0.07	0.22	−0.10	−0.24
Walk Time	0.42	−0.02	−0.27	0.03	−0.36	−0.42	0.19	−0.11	−0.21

^ Sensitivity refers to power analysis indicating the minimum correlation likely to be detectable at an alpha of 0.1 and a given n. ~0.3 refers to the equivalent medium correlation for general linear models with f2 scores of 0.16 and 0.16 which is a medium effect size [30]. ^†^
*p* < 0.10, **p* < 0.05, ** *p* < 0.01.

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
