# Peer review of "Grazing Cow Behavior’s Association with Mild and Moderate Lameness"

_animals, 2020, doi:10.3390/ani10040661_

Round 1
Reviewer 1 Report
The Authors have deeply revised their paper not only addressing our remarks regarding the methods but also adding appropriate considerations in the discussion as well in the introduction paragraphs.
Just two little remarks:
- it seems that the (important) sentence at row 98-99 lacks the question mark.
- Please rephrase sentence at row 115-116
Author Response
Just two little remarks:
- it seems that the (important) sentence at row 98-99 lacks the question mark.
- Please rephrase sentence at row 115-116
These have been implemented.
Reviewer 2 Report
I prefer this revised version to the previous one. The authors respond appropriately to my previous questions. However, the authors extensively revised the manuscript; thus, I find several small issued as show below:
- In the response letter, the authors asked me that “Is the reviewer referring to lameness prevalence in grazing systems? Or the difficulty of scoring cows at pasture?”
The answer is both. My question is why the authors targeted "grazing cows" in this study. In my speculation, lameness is more prevalent in non-grazing cows than in grazing cows; thus, most of the past studies focused on zero-grazing (housed) cows. However, this study focused on "grazing cows". So,
1) Are grazing cows more likely to have lameness than non-grazing cows? or is lameness a problem in grazing cows as is the case in non-grazing cows? Showing lameness prevalence may be helpful to support your idea.
2) Is there any difficulty to detect lameness in grazing cows?
The authors have responded some of my question in the revised version, but I recommend that you should add stronger reason(s).
- I am a bit unsure whether scoring events ‘A’ were implemented at the beginning and events ‘B’ were implemented at the end of trials in trials 1, 3 and 4. The writing in Table 1 sounds a bit confusing.
If this is not true, is there any rational reason to combine these data sets, i.e., 1A, 2, 3A and 4A and 1B, 2, 3B and 4B? I mean you could choose any combination of the data set; for example, you can combine 1A, 2, 3B and 4A into the aggregated data.
Details:
Line 31: Replace "17 good mobility" with "Seventeen good mobility".
Line 126: Accuracy of the RumiWatch to detect those 14 behaviors should be discussed by other reported studies.
Line 139: The results of power analysis should be shown in Table 3 (or Table 4).
Line 141: Describe structure of this generalized linear model clearly. What are the factors?
Author Response
- In the response letter, the authors asked me that “Is the reviewer referring to lameness prevalence in grazing systems? Or the difficulty of scoring cows at pasture?”
The answer is both. My question is why the authors targeted "grazing cows" in this study. In my speculation, lameness is more prevalent in non-grazing cows than in grazing cows; thus, most of the past studies focused on zero-grazing (housed) cows. However, this study focused on "grazing cows". So,
1) Are grazing cows more likely to have lameness than non-grazing cows? or is lameness a problem in grazing cows as is the case in non-grazing cows? Showing lameness prevalence may be helpful to support your idea.
These are important questions which are tangentially relevant to the paper. Grazing cows tend to be less likely to be lame than housed cows I believe – in agreement the reviewers estimation. However, lameness is still a major issue for grazing animals and we now support this point with references LN 53.
2) Is there any difficulty to detect lameness in grazing cows?
The process of manual locomotion scoring in grazing cows is to the authors knowledge similar / identical to zero grazing cows. If automated lameness detection is harder for grazing cows is unknown as the capability is not well proven by anyone in either grazing or zero grazing contexts. But automated lameness detection is as important in both settings.
The authors have responded some of my question in the revised version, but I recommend that you should add stronger reason(s).
- I am a bit unsure whether scoring events ‘A’ were implemented at the beginning and events ‘B’ were implemented at the end of trials in trials 1, 3 and 4. The writing in Table 1 sounds a bit confusing.
The table 1 text has been revised. A is for the beginning and B is for the end of the trials and this has been revised in the text. LN119-124.
If this is not true, is there any rational reason to combine these data sets, i.e., 1A, 2, 3A and 4A and 1B, 2, 3B and 4B? I mean you could choose any combination of the data set; for example, you can combine 1A, 2, 3B and 4A into the aggregated data.
Yes – the choice is arbitrary but this seems the most logical.
Details:
Line 31: Replace "17 good mobility" with "Seventeen good mobility".
Implemented.
Line 126: Accuracy of the RumiWatch to detect those 14 behaviors should be discussed by other reported studies.
This is now discussed LN 127:130.
Line 139: The results of power analysis should be shown in Table 3 (or Table 4).
Thank for you this suggestion, We have revised the presentation of the power analysis and include it in Table 4.
Line 141: Describe structure of this generalized linear model clearly. What are the factors?
This has been implemented.
This manuscript is a resubmission of an earlier submission. The following is a list of the peer review reports and author responses from that submission.
Round 1
Reviewer 1 Report
The Authors are invited to provide the inclusion criteria and the sampling method for the selection of the animals to be monitored.
Furthermore, it should be noted that the total number of 65 animals doesn’t correspond to the sum of the animals in each trial.
The lack of statistically significant correlations might be due not only to little sample size but also to low prevalence of mild or moderate lameness.
The results of the lameness (mobility) scoring in each trial should be indicated.
Consequently, the Authors are invited to compute the achieved power of the statistic test adopted in their study having regard of the sample size of each trial (n. 19-11) and the small effect size (i.e. lameness) they have wanted to show.
The gap between the scoring and the monitoring could lead to a failure in the correlation between the manual score and the behavioral measurements considering that mild and moderate lameness can have a rapid onset and/or resolution.
I wasn’t able to find the following references:
5 (http://www.icerobotics.com/news/cowalert-lameness-detection-highly-commended/) 17 (https://dairy.ahdb.org.uk/resources-library/technical-information/health-welfare/mobility-score- instructions/#.XiUM0cgzYdU)The reference cited at row 153 (Blackie et al, 2011) is not reported in the References section
The reference [31] appears inappropriate because the review it refers to is yet to be published, therefore this statement is not verifiable.
Author Response
Revised manuscript attached. Below - letter to reviewers and below that, responses to specific queries.
Letter to the reviewers following first round of comments
Dear Reviewers,
We would like to thank you for your thoughtful, thorough, robust and very constructive input. We have revised the paper significantly in light of these comments including the reporting of additional data, power analysis and analysis of aggregated data. We have rewritten most of the paper and so do not mark changes as this would be mark than 70% of the paper. In particular, we discuss in more detail the role of grazing / zero grazing, severe/ non-severe lameness and breed. Upon further reflection, we also clarify that the papers’ value is not in discerning if associations exist in controlled conditions – but rather asking
‘are the studied behaviors likely to be useful for automated lameness detection in the studied commercially relevant contexts?’
Though the main findings of the initial submission remain broadly similar there are some changes. In performing the additional analysis requested by reviewer 1, 2 coding errors were identified and are now rectified. The first of these was an error in trial one matching cows to recorded behaviour, and the other in later trials which excluded some cows incorrectly. Some other cows are now excluded as a closer inspection of the data identified unfeasibly low walking time data. As such the sample sizes of the individual trials reported has changed (16 +16+ 16+15 = 63). With additional aggregated data analysis we now report a significant association between standing up/laying counter activity and lameness.
We have also clarified the strengths and limitations of the reported studies. We wish to highlight here the strengths - multiple trials, multiple breeds, multiple farms, open access data and statistical analysis code. In the discussion, a limitation section now includes a discussion of parity and acknowledges this study would ideally have had greater statistical power. Reviewer 2 recommended accounting for / controlling for a range of variables including parity, distance to paddock and weather. Of these, parity has been identified as a confounder and it could be integrated into a commercial system relatively easily. However, we did not find any associations with behaviour and lameness in our study. Regarding weather, precipitation is the main concern in the literture and this was minimal for the studied days and so did not affect this study.
For the other variables e.g. distance to paddock or feed allowance - it would be a challenge to include these in a commercially viable automated system. We would thus see these variables as necessarily being non-essential/optional for a commercially viable lameness detection system. If a behaviour is likely to be useful for automated detection in pasture based systems, there should be statistically robust correlations without having to control for several variables. Controlling for variables should be seen as a way to increase performance – not essential to discerning any statistically robust difference.
With the possible exception of standing up/laying counter and activity, we still believe the results indicate the studied variables are likely to be of limited use for lameness detection in the studied context. Though not reported in the paper, we also analysed the following and found similar results to that with the basic analysis.
- behaviour between milkings,
- ratio of day to night time behaviour,
- change in behaviour and locomotion for 3 trials, and,
- measures of variance.
We have published the code and data required to run this analysis on our github for reviewers interested in running this analysis themselves and viewing the results. https://github.com/nialloleary/Behavior-Lameness-Article
Based on our initial reading of the positive sounding literature, we were disappointed we did not find positive results. This studies findings are, however, congruent with this review some of the authors recently contributed to. (https://www.sciencedirect.com/science/article/pii/S0022030220301417?via%3Dihub). Progress by focusing on behaviour has been limited and we believe this is in large part because of a failure to replicate and failure to acknowledge those findings reported are relatively modest in effect size and inconsistent. In the review, we conclude behaviour should no longer be the main focus. We further argue that analysis of gait should be the focus of further research (e.g. Halahchen et al, Alsaaod et al) and a combination of gait and behaviour, with a focus on gait is the most promising avenue for accelerometer based lameness detection.
All 3 reviewers engaged constructively and thoughtfully with this paper. Thank you again for your excellent contributions and we look forward to any further remarks you may have. We hope we have address most, if not all those raised so far to your satisfaction.
Regards,
Niall O’Leary
Bold text indicates author responses.
The Authors are invited to provide the inclusion criteria and the sampling method for the selection of the animals to be monitored.
“One experienced technician who trains other scorers, mobility scored cows as they exited the milking parlor. Mildly and moderately lame cows were selected on a first scored, first selected basis while approximately every third cow selected was nonlame to ensure non lame cows had comparable milking order” LN120:123
Furthermore, it should be noted that the total number of 65 animals doesn’t correspond to the sum of the animals in each trial.
Thank you for highlighting this inconsistency. Revised text “On a research farm and a commercial farm, 63 grazing cows in total were studied across 4 trials (16, 16, 16 and 15 cows).’ LN 31:32.
The lack of statistically significant correlations might be due not only to little sample size but also to low prevalence of mild or moderate lameness.
The samples were stratified by mobility score, and across trials there were similar numbers of nonlame, mildly lame and moderately lame cows.
“ Based on visual observation of cows by a trained mobility scorer on a 4-point scale, 17 nonlame (score 0), 23 mildly lame (score 1) and 22 moderately lame (score 2) cows were selected for monitoring using leg worn accelerometers.” LN32-34
The results of the lameness (mobility) scoring in each trial should be indicated.
Mobility scoring stopped after 6/7 mildly lame cows and 6/7 moderately lame cows were identified. Therefore, herd mobility statistics were not collected. The mobility score prevalence’s of tracked cows are now reported in Table 1
Consequently, the Authors are invited to compute the achieved power of the statistic test adopted in their study having regard of the sample size of each trial (n. 19-11) and the small effect size (i.e. lameness) they have wanted to show.
Power analysis is now reported in the results. LN 162-171. Related to this, 4 datasets are merged to have to create a pooled analysis. LN 171.
The gap between the scoring and the monitoring could lead to a failure in the correlation between the manual score and the behavioural measurements considering that mild and moderate lameness can have a rapid onset and/or resolution.
We have now included the following in “The gaps between scoring and behavior measurements (table 1) could also have coincided with changes in mobility score that would have obscured associations.” LN 232
I wasn’t able to find the following references:
5 (http://www.icerobotics.com/news/cowalert-lameness-detection-highly-commended/) 17 (https://dairy.ahdb.org.uk/resources-library/technical-information/health-welfare/mobility-score- instructions/#.XiUM0cgzYdU).The reference cited at row 153 (Blackie et al, 2011) is not reported in the References section.
These have been remedied.
The reference [31] appears inappropriate because the review it refers to is yet to be published, therefore this statement is not verifiable.
This article has since been published. https://www.sciencedirect.com/science/article/pii/S0022030220301417?via%3Dihub

Reviewer 2 Report
General comments
This research examined the relationships between a mobility score (which is related to lameness score) and activity measured using a pedometer in a pasture-based system. A total of 65 cows were studied over a total of 4 farms.
While I appreciate what the authors are attempting to do with this paper, I do not think it has appropriately framed by the introduction, I have questions about the methodology, and there are additional results that need to be presented. My more significant comments are detailed below, with further comments with specific line numbers after that. Based on these comments, I unfortunately cannot recommend this paper for publication in its present format. The paper could be improved with the inclusion of additional data, a better focused introduction and discussion that contrasts findings in pasture-based to indoor systems, and a better recognition of the limitations of the research given the methodology.
Firstly, the point of difference for this paper is the examination of milk/moderate lameness in a pasture based system. This point of difference needs to be better established in the introduction. At present, the conflicting results in relationships between behaviour and lameness are discussed, but is unclear if this refers to pasture based or indoor systems. It is also not clear that the authors are assessing mild/moderate lameness, where this sits in the context of previous research, and the importance of these categories of lameness in terms of intervention. The same can be said of the discussion, a better compare/contrast to previous findings is required.
Secondly, the mobility assessments were not accompanied with a clinical assessment of lameness, so we cannot be sure that these animals were actually lame. Further, there are a number of external factors that could affect activity, and these relationships are not examined. In particular, the cows had to walk a distance to and from the dairy, regardless of their mobility score. I wonder if the authors restricted their behavioural observations only to the time cows were at pasture if there would be a difference in the findings. Also, animals naturally vary in their activity levels and so a “change” from normal activity levels is more suitable than a correlation assessment at each observation day.
Finally, there are no reports on the number of animals in each category of mobility, and these may not be proportionately relatable to a larger herd. Thus, on a larger herd where more 0 scores are expected different outcomes may be seen.
Specific comments
Title: This title doesn’t fully captures what the article is about. Maybe something like “Associations between cow activity and lameness in grazing dairy cattle”?
Simple summary: I think the simple summary would benefit from some addition details. The strengths of this research is (1) replication and (2) grazing cattle, but neither of these are emphasised well in the summary. Some suggested changes are listed below.
L14: Cow behaviour – what behaviours? Activity?
L14: Lameness(mobility) - What does this mean? Do you mean lameness or mobility? If you are using one as a proxy for the other you need to make that clear. At L18 you just refer to lameness? Same for L19 of the abstract
L15: “most promising reported associations” – what makes them the most promising? Identified from previous research on indoor cattle?
L: Delete “different” before “farms”, it is a given that the two farms are different!
L16: Delete “However” at the beginning of the sentence, it interrupts the flow of the writing
L17: What were the behaviours studies? If too many to specifically state, you should mention that they are related to activity (‘behaviour’ is very broad)
L17: “…were found not to be associated consistently…” This is tricky wording. Suggest something like “There were no consistent associations between the behaviours studied and milk or moderate lameness”
L18: “utility” – do you mean “usefulness”?
Abstract
L19 and 23: Please make it clear whether you assessed lameness or mobility
L24: 65 cows in total? How many at each farm/trial?
L24-25: How was mobility assessed (how points on the scale, how many observers? How many cows in each category?
Introduction
General comment: I think the interesting aspect of this research is that it has conducted these observations on animals with mild to moderate lameness in grazing systems. However, these points of difference are not introduced very well. It would be good if the authors tease out the references provided in the last paragraph – ie of the papers referenced, how many were conducted on grazing cattle, and, if any, how did the associations between lameness and activity differ to those in indoor systems? Similarly for the lameness/mobility scale, the authors are focussing on mild/moderate – what were the scales utilised by the research with high associations?
L32: Are you speaking generally, or do you mean in dairy cattle?
L33-34: Again, what does lameness(mobility) mean? If you used one as a measure of the other, you need to make there relationship clear, with references if possible.
L35: “…estimate of total lameness” – needs a reference
L47: “…changes in behaviour”: Please provide some more specific details of behaviour. What aspects have been associated with lameness? For example, walking duration, motion index and lying bouts could be considered as changes in behaviour in themselves
Materials and methods
General comment: The materials and methods follows an unusual structure. Typically, these start with details on the animals and farms eg animals used each farm, breed, how long were they observed, farm details, herd size etc. I can see this is summarised in Table 1, but needs to introduced up front. Details on scoring should follow this and then the details on the behaviour monitoring. At the moment this section jumps all over the place – from scoring, to pedometers, to animals and housing, back to pedometers, back to scoring and them finally back to pedometers. This could even be presented as different subsections “Animal and management” “Mobility scoring” “Activity monitoring”
L56: I think the first sentence here is out of place. If anything, it belongs in the introduction along with a definition of mobility and evidence of the association between mobility and lameness
L57: the numeral 0 is used at the start of a sentence, it needs to be “Zero”
L57-58: More details are required on these scales. What constitutes “imperfect mobility”? If someone where to repeat this study, what behaviours would thy need to see in order for to be categorised as so? Same for all classifications – an ethogram is probably required.
L59-64: “Horseman…of scoring the cows” – this all belongs in the introduction I think
L65: More details are required on the observer. Were they experienced/trained observer? A researcher or a farm staff? Who trained them? This research woulod have been strengthened if the animals were also clinically assessed to assure confidence in the observers scoring. I understand you can’t go back and do this, but this is a limitation and needs to be discussed. Also, deleted the “,” between “One scorer” and “mobility scored”
L72-73: “Sample size…multiple times”. I don’t think this is necessary.
L75: Please list numbers in each trial or refer to Table 1 here
L76: What sort of technical failures? How long did you have the pedometers on for? Failure of 10 pedometers is nearly 50% failure rate – how confident are you with the remainders? Were there failures in the other trials too?
L77-78: Here you talk about initial and final scoring days but you haven’t introduced this well. Please make this more clear. Were they scored every day between the initial and final or just at the initial and at the final? Restructuring this section as suggested earlier may be helpful
L78-80: How early in lactation were the cows? What was there parity (mean and range)? How much supplementation (range and mean) were they provided? Was this as concentrate in the dairy? How much pasture were they offered? Did cows remain in the herd or were they managed as a separate group? This would have implications for waiting time at the dairy which affect activity levels. What were the weather conditions when you scored (these details could be presented as another subsection – can effect behaviour as bad weather = reduced grazing so reduced steps, motion index etc). I also think you should provide the mean and range of distances to the dairy. As you are recording activity, I would say this is a very important factor to report, and may even be a contributing factor to why you failed to find associations.
L82: “Behaviour summaries”, you need to be specific about the behaviours recorded in the methods
L84: Why only scored once in trial 2?
L85: “24 h data” – given that all cows in a group had to walk the same distance to and from the dairy each day, regardless of mobility score, I wonder if assessing differences in behaviour only for the time they are at pasture would be appropriate? Perhaps the distance travelled is “washing out” effects observed while the animal is “at leisure”? I understand that this would not be useful to a farmer, but it might be useful in determining variability in relationships in the literature as well as the applicability of changes in behaviour as an indicator of lameness for cattle at pasture.
L86: Do you mean Table 2 (rather than Table3)?
L86-87: “Spearmans Rho…variables and mobility score” – Separately for each day? Or as a total? For each farm? Or were all scores on all farms totalled? Is this the most appropriate way of handling categorical data (mobility score)? Why not use a non-parametric test such as Kruskal-Wallis? You could even report on the results of both. Would you expect differences depending on the farm, distance walked, weather, how long they had to wait at the dairy (time off pasture) etc? You could also analyse and report such relationships, I would be more confident in a correlation or KW if you assessed the significance of other possible contributions, and given the limited data presented in this research it could make the manuscript more substantial. Further, wouldn’t a change from normal for individual cows be a more useful/practical indication of lameness ie you might have a cow that is generally low in activity that might not be lame, or you could have a cow that is usually very active suddenly drop in activity, which would be a more practical indication the something is wrong with that animal. This is partly why a clinical assessment of cows identified as lame would have strengthened this study.
Table 1: Please include a footnote where A and B are defined. A lot of the detail in this table (especially the last column) would be better described in the methods.
Table 2. References should be in number format. Strides, Walking counter, Walking index, Walking time – all examples of variable that would be affected by distance to and from the dairy (which all animals have to do, regardless of mobility), DM offered, weather etc.
Results
Generally, I think more data needs to presented in the results. At the very least, there is no data on the number of cows in each mobility classification and this must be included. I would also like to see the authors explore the relationships between activity and mobility (1) as a Kruskal wallis non-parametric test, and (2) for the time that cows were at pasture. External variables (farm, kg pasture DM allocated, weather conditions, time off pasture, distance to the dairy) should also be investigated and eliminated.
L103-104: “This was…(distance to paddock)” This is a discussion point and not a result
L112-114: “In summary…walking time”. This is also a discussion and not a result
Table 4: Why are there so many missing data in this Table?
Discussion
L126: delete “in this study” at the end of the sentence
L127: are the references provided here for cows at pasture? It would be good to compare/contrast these findings for cows at pasture to similar experiments on cows housed indoors.
L129-131: Or, there are things you haven’t controlled for that are washing out the relationships?
L133: I think the key is to examine changes in behaviour for individual animals, rather than a blanket behavioural assessment.
L134-135: sample sizes – these are small sample sizes, and you haven’t provided numbers for the cows within each mobility category you used or on whether the animals studied were randomly chosen or chosen because they were identified as having compromised mobility. This makes it hard to apply your findings to a large herd. For example, on a large herd of cattle at pasture it would be expected that the vast majority of animals would have a mobility score of 0. In differences in behaviour for a cow with a mobility score of 1 or 2 and definitely 3 may be statistically different. But in your studied herds the proportion of animals with impaired mobility and those without I presume (but do not know because the data isn’t presented) would not be representative of a commercial farm. This is why a change in behaviour may be more reliable.
L137-138: Again, hard for the reader to assess this as numbers of cows in each category were not reported
L138-139: Here also, I make the point regarding proportion of animals in each category in the studied group compared to a larger herd and of changes in behaviour perhaps being more appropriate.
L142-146: I think this is an important paragraph and this is why I think you should compare to relationships between behaviour restricted to the times only when cows are at pasture. What do the experiments for cows at pasture report? Are they different to what you have reported? This is also perhaps where a discussion on the usefulness on “changes from normal” behaviour for individual animals.
L149: Ideally you would’ve confirmed the reliability of the observers assessment firstly with a clinical assessment of the animal, and a secondary observer.
L150: This is the first time the authors have said that severely lame animals were not included in the study. Was this deliberate? Ideally in a pasture based systems severely lame animals will be moved to sick herd close to the dairy so its usefulness for this cohort of animals is questionable.
L153: And what did Blackie find? The same as the present research? Did Blackie study animals at pasture?
L161-163: This sentence is repetitive of early paragraphs
L165-166: I would like to see the authors expand on this discussion point, as it is important. Have no behaviours been found useful in the non-grazing animal? Or some behaviours? How many studies have found no association? Have any reported contrary results? If there truly are no relationships for mild/moderate lameness in either system, then why? Is the animals behaviour just not affected enough by the condition at this early stage?
L168-169: I think you can delete the first sentence of this paragraph.
Author Response
Please find attached a revised version of the manuscript. Below a letter to the reviewers. Below that, responses to specific reviewer queries.
Letter to the reviewers following first round of comments
Dear Reviewers,
We would like to thank you for your thoughtful, thorough, robust and very constructive input. We have revised the paper significantly in light of these comments including the reporting of additional data, power analysis and analysis of aggregated data. We have rewritten most of the paper and so do not mark changes as this would be mark than 70% of the paper. In particular, we discuss in more detail the role of grazing / zero grazing, severe/ non-severe lameness and breed. Upon further reflection, we also clarify that the papers’ value is not in discerning if associations exist in controlled conditions – but rather asking
‘are the studied behaviors likely to be useful for automated lameness detection in the studied commercially relevant contexts?’
Though the main findings of the initial submission remain broadly similar there are some changes. In performing the additional analysis requested by reviewer 1, 2 coding errors were identified and are now rectified. The first of these was an error in trial one matching cows to recorded behaviour, and the other in later trials which excluded some cows incorrectly. Some other cows are now excluded as a closer inspection of the data identified unfeasibly low walking time data. As such the sample sizes of the individual trials reported has changed (16 +16+ 16+15 = 63). With additional aggregated data analysis we now report a significant association between standing up/laying counter activity and lameness.
We have also clarified the strengths and limitations of the reported studies. We wish to highlight here the strengths - multiple trials, multiple breeds, multiple farms, open access data and statistical analysis code. In the discussion, a limitation section now includes a discussion of parity and acknowledges this study would ideally have had greater statistical power. Reviewer 2 recommended accounting for / controlling for a range of variables including parity, distance to paddock and weather. Of these, parity has been identified as a confounder and it could be integrated into a commercial system relatively easily. However, we did not find any associations with behaviour and lameness in our study. Regarding weather, precipitation is the main concern in the literture and this was minimal for the studied days and so did not affect this study.
For the other variables e.g. distance to paddock or feed allowance - it would be a challenge to include these in a commercially viable automated system. We would thus see these variables as necessarily being non-essential/optional for a commercially viable lameness detection system. If a behaviour is likely to be useful for automated detection in pasture based systems, there should be statistically robust correlations without having to control for several variables. Controlling for variables should be seen as a way to increase performance – not essential to discerning any statistically robust difference.
With the possible exception of standing up/laying counter and activity, we still believe the results indicate the studied variables are likely to be of limited use for lameness detection in the studied context. Though not reported in the paper, we also analysed the following and found similar results to that with the basic analysis.
- behaviour between milkings,
- ratio of day to night time behaviour,
- change in behaviour and locomotion for 3 trials, and,
- measures of variance.
We have published the code and data required to run this analysis on our github for reviewers interested in running this analysis themselves and viewing the results. https://github.com/nialloleary/Behavior-Lameness-Article
Based on our initial reading of the positive sounding literature, we were disappointed we did not find positive results. This studies findings are, however, congruent with this review some of the authors recently contributed to. (https://www.sciencedirect.com/science/article/pii/S0022030220301417?via%3Dihub). Progress by focusing on behaviour has been limited and we believe this is in large part because of a failure to replicate and failure to acknowledge those findings reported are relatively modest in effect size and inconsistent. In the review, we conclude behaviour should no longer be the main focus. We further argue that analysis of gait should be the focus of further research (e.g. Halahchen et al, Alsaaod et al) and a combination of gait and behaviour, with a focus on gait is the most promising avenue for accelerometer based lameness detection.
All 3 reviewers engaged constructively and thoughtfully with this paper. Thank you again for your excellent contributions and we look forward to any further remarks you may have. We hope we have address most, if not all those raised so far to your satisfaction.
Regards,
Niall O’Leary
General comments
This research examined the relationships between a mobility score (which is related to lameness score) and activity measured using a pedometer in a pasture-based system. A total of 65 cows were studied over a total of 4 farms.
While I appreciate what the authors are attempting to do with this paper, I do not think it has appropriately framed by the introduction, I have questions about the methodology, and there are additional results that need to be presented. My more significant comments are detailed below, with further comments with specific line numbers after that. Based on these comments, I unfortunately cannot recommend this paper for publication in its present format. The paper could be improved with the inclusion of additional data, a better focused introduction and discussion that contrasts findings in pasture-based to indoor systems, and a better recognition of the limitations of the research given the methodology.
Firstly, the point of difference for this paper is the examination of mild/moderate lameness in a pasture based system. This point of difference needs to be better established in the introduction. At present, the conflicting results in relationships between behaviour and lameness are discussed, but is unclear if this refers to pasture based or indoor systems. It is also not clear that the authors are assessing mild/moderate lameness, where this sits in the context of previous research, and the importance of these categories of lameness in terms of intervention. The same can be said of the discussion, a better compare/contrast to previous findings is required.
The introduction has been revised addressing these comments. “Much has varied between studies of behavior-lameness associations. In addition to the specific behaviors examined and if the cows were grazing or not, the severity of lameness examined in studies and the breed of animal studied has varied …”LN 83-84 continued to 95.
Secondly, the mobility assessments were not accompanied with a clinical assessment of lameness, so we cannot be sure that these animals were actually lame.
We acknowledge this point and now include hoof inspections as a suggested improvement to the methodology LN 230. However, the focus here is automated mobility scoring as opposed to an automated proxy for hoof inspections. Hoof inspections were carried out (not reported in the paper) during the first trial of Jerseys grading each hoof for overgrown, sole haemorrhage and white line disease. These inspections confirmed the presence of issues on each hoof of each score 2 cow with White Line disease being the most prevalent. Each score 1 cow also had issues (usually white line) on each hoof, though they were less severe. Some indications of mild white line disease were also found for most hoofs of cows of score 0. Overall, hoof inspections indicated differences in the prevalence of pathologies somewhat consistent with mobility score but did not appear to add significant value and so were not continued for the remaining trials.
Further, there are a number of external factors that could affect activity, and these relationships are not examined. In particular, the cows had to walk a distance to and from the dairy, regardless of their mobility score. I wonder if the authors restricted their behavioural observations only to the time cows were at pasture if there would be a difference in the findings.
This is a very valid point which the authors believe undermines the overall potential for using behaviour to detect lameness on commercial farms. Even if it is possible in controlled conditions, it is only relevant if it can be applied on commercial farms. This point would be a major challenge for commercial application. However, analysis was done looking at this between milking time and similar modest results were found. For brevity, they are not reported in this paper. This additional analysis and raw data has been made available through github (https://github.com/nialloleary/Behavior-Lameness-Article).
Also, animals naturally vary in their activity levels and so a “change” from normal activity levels is more suitable than a correlation assessment at each observation day.
This point has some support (Byabazaire et al., 2019) however, good predictive validity has been achieved by observing just 2 days (Beer et al., 2016). Analysis of change in behavior was also performed for 3 trials and similar weak results were found and are not reported. The data and supplementary analysis scripts are however available via the github repo (https://github.com/nialloleary/Behavior-Lameness-Article).
Finally, there are no reports on the number of animals in each category of mobility, and these may not be proportionately relatable to a larger herd. Thus, on a larger herd where more 0 scores are expected different outcomes may be seen.
Table 1 now contains numbers of cows by each score for each trial. Across the trials, the numbers were similar.
Specific comments
Title: This title doesn’t fully captures what the article is about. Maybe something like “Associations between cow activity and lameness in grazing dairy cattle”?
Thank you for your suggestion. We have revised the title.
Simple summary: I think the simple summary would benefit from some addition details. The strengths of this research is (1) replication and (2) grazing cattle, but neither of these are emphasised well in the summary. Some suggested changes are listed below. L14: Cow behaviour – what behaviours? Activity? L14: Lameness (mobility) - What does this mean? Do you mean lameness or mobility? If you are using one as a proxy for the other you need to make that clear. At L18 you just refer to lameness? Same for L19 of the abstract L15: “most promising reported associations” – what makes them the most promising? Identified from previous research on indoor cattle? Delete “different” before “farms”, it is a given that the two farms are different! L16: Delete “However” at the beginning of the sentence, it interrupts the flow of the writing. L17: What were the behaviours studies? If too many to specifically state, you should mention that they are related to activity (‘behaviour’ is very broad). L17: “…were found not to be associated consistently…” This is tricky wording. Suggest something like “There were no consistent associations between the behaviours studied and milk or moderate lameness”. L18: “utility” – do you mean “usefulness”?
The simple summary has been revised addressing these comments.
Abstract
L19 and 23: Please make it clear whether you assessed lameness or mobility
This has been clarified.
L24: 65 cows in total? How many at each farm/trial?
This has been specified.
L24-25: How was mobility assessed (how points on the scale, how many observers? How many cows in each category?
This is now described in ln ---117-124 and Table 1.
Introduction
General comment: I think the interesting aspect of this research is that it has conducted these observations on animals with mild to moderate lameness in grazing systems. However, these points of difference are not introduced very well. It would be good if the authors tease out the references provided in the last paragraph – ie of the papers referenced, how many were conducted on grazing cattle, and, if any, how did the associations between lameness and activity differ to those in indoor systems? Similarly for the lameness/mobility scale, the authors are focussing on mild/moderate – what were the scales utilised by the research with high associations?
These observations have informed a revised introduction which now focuses on pasture based literature more.
L32: Are you speaking generally, or do you mean in dairy cattle?
Dairy cattle – dairy has been specified. LN54
L33-34: Again, what does lameness (mobility) mean? If you used one as a measure of the other, you need to make their relationship clear, with references if possible.
There is now a detailed discussion of this. LN 43-53. , 128-131
L35: “…estimate of total lameness” – needs a reference
Reference added. Ln57
L47: “…changes in behaviour”: Please provide some more specific details of behaviour. What aspects have been associated with lameness? For example, walking duration, motion index and lying bouts could be considered as changes in behaviour in themselves
This has now been revised LN69-82
Materials and methods
General comment: The materials and methods follows an unusual structure. Typically, these start with details on the animals and farms eg animals used each farm, breed, how long were they observed, farm details, herd size etc. I can see this is summarised in Table 1, but needs to introduced up front. Details on scoring should follow this and then the details on the behaviour monitoring. At the moment this section jumps all over the place – from scoring, to pedometers, to animals and housing, back to pedometers, back to scoring and them finally back to pedometers. This could even be presented as different subsections “Animal and management” “Mobility scoring” “Activity monitoring”
This section has been reorganised.
L56: I think the first sentence here is out of place. If anything, it belongs in the introduction along with a definition of mobility and evidence of the association between mobility and lameness
This discussion has been revised and moved to the introduction. LN 43:53.
L57: the numeral 0 is used at the start of a sentence, it needs to be “Zero”
Revised.
L57-58: More details are required on these scales. What constitutes “imperfect mobility”? If someone where to repeat this study, what behaviours would thy need to see in order for to be categorised as so? Same for all classifications – an ethogram is probably required.
We believe the reference to the AHDB scoring system used is sufficient for replication purposes.
L59-64: “Horseman…of scoring the cows” – this all belongs in the introduction I think
Moved to introduction.
L65: More details are required on the observer. Were they experienced/trained observer? A researcher or a farm staff? Who trained them? This research would have been strengthened if the animals were also clinically assessed to assure confidence in the observers scoring. I understand you can’t go back and do this, but this is a limitation and needs to be discussed. Also, deleted the “,” between “One scorer” and “mobility scored”.
This has been revised as follows ‘One experienced technician who has trained other mobility scorers, mobility scored cows as they exited the milking parlor’ LN120:125. The scorer was a senior technician with decades of experience of locomotion scorings who has since retired. He was one of the scorers in this trial for example https://www.sciencedirect.com/science/article/pii/S0022030218308531.
Here is a video of the scorer, Noel Byrne, providing locomotion scoring training to others prior to the data collection period https://www.youtube.com/watch?v=JYi316YeIJg.
Hoof inspections were carried out during the first trial of Jerseys grading each hoof for overgrown, sole haemorrhage and white line disease. These inspections confirmed the presence of issues on each hoof of each score 2 cow with White Line disease being the most prevalent. Each score 1 cow also had issues (usually white line) on each hoof, though they were less severe. Some indications of mild white line disease were also found for most hoofs of cows of score 0. Overall, hoof inspections indicated differences in the prevalence of pathologies somewhat consistent with mobility score.
L72-73: “Sample size…multiple times”. I don’t think this is necessary.
Edited.
L75: Please list numbers in each trial or refer to Table 1 here
This discussion has been revised as has been Table 1.
L76: What sort of technical failures? How long did you have the pedometers on for? Failure of 10 pedometers is nearly 50% failure rate – how confident are you with the remainders? Were there failures in the other trials too?
“Operator errors and technical faults resulted in one of the pedometers failing to operate at all, one appearing to operate but failing to record data and three recording erroneous data with no or unfeasibly low walking time. These latter data sets were excluded from the analysis.” LN 137
The main issue we had was recording the data and finding some pedometers did not record walking (so needed to be excluded). Our inspection of the remaining data indicates the analysed data is reliable. Furthermore, we are publishing this data for others to inspect and use.
L77-78: Here you talk about initial and final scoring days but you haven’t introduced this well. Please make this more clear. Were they scored every day between the initial and final or just at the initial and at the final? Restructuring this section as suggested earlier may be helpful
This has been revised. LN128:129.
L78-80: How early in lactation were the cows?
LN112 – 115. All cows were spring calved in January/February/ March and scored between June and August and so would have been mid lactation and their behaviour should not have been affected by estrus.
What was there parity (mean and range)?
This is now described LN141:144.
How much supplementation (range and mean) were they provided? Was this as concentrate in the dairy?
0-2kgs per head per day. Concentrate was provided in the parlor. LN112.
How much pasture were they offered?
This information was not aggregated and it would be hard to integrate into a commercial product.
Did cows remain in the herd or were they managed as a separate group? This would have implications for waiting time at the dairy which affect activity levels.
They remained in the herd of group of 40 to 200 cows. LN 124.
What were the weather conditions when you scored (these details could be presented as another subsection – can effect behaviour as bad weather = reduced grazing so reduced steps, motion index etc).
Consulting historical records, we see that weather was good during the trials. Trial 1.a had 0.7mm of rain on the day of observation and 1.b 0.1mm. The rest were dry. LN114.
I also think you should provide the mean and range of distances to the dairy. As you are recording activity, I would say this is a very important factor to report, and may even be a contributing factor to why you failed to find associations.
This data was not collected and distance to paddock would be a very challenging variable to integrate into a commercial product.
L82: “Behaviour summaries”, you need to be specific about the behaviours recorded in the methods
Now states 14 behaviors and references Table 2.
L84: Why only scored once in trial 2?
In trial 2, cows were only scored once due to locomotion scorer unavailability. LN130:132
L85: “24 h data” – given that all cows in a group had to walk the same distance to and from the dairy each day, regardless of mobility score, I wonder if assessing differences in behaviour only for the time they are at pasture would be appropriate? Perhaps the distance travelled is “washing out” effects observed while the animal is “at leisure”? I understand that this would not be useful to a farmer, but it might be useful in determining variability in relationships in the literature as well as the applicability of changes in behaviour as an indicator of lameness for cattle at pasture.
We had similar thoughts. We considered including a section on variation in behaviour through out the day. This analysis was run (’6 additional analysis Day Night’) excluding milking and transit time. However, there were similarly modest results and we decided to exclude this analysis for brevity and conciseness.
L86: Do you mean Table 2 (rather than Table3)?
Table 3 is correct.
L86-87: “Spearmans Rho…variables and mobility score” – Separately for each day? Or as a total? For each farm? Or were all scores on all farms totalled?
They were reported in table 4 for each locomotion scoring event. We have now aggregated four locomotion scoring events, one form each trial, and presented additional Linear model analysis where trial is a variable.
Is this the most appropriate way of handling categorical data (mobility score)? Why not use a non-parametric test such as Kruskal Wallis? You could even report on the results of both.
It is our understanding that Spearman’s rank is an appropriate non parametric analysis for ordinal data (like locomotion score). Kruskall Wallace would be more appropriate for categorical and ordinal data. For example, Kruskall Wallace would be useful for assessing lying time between cows with white line, sole haemorrhages and digital dermatitis.
Would you expect differences depending on the farm, distance walked, weather, how long they had to wait at the dairy (time off pasture) etc?
Yes, but we contend that practical relevance requires associations to still be observable without controlling for that which is unlikely to be controllable for in a practical application.
You could also analyse and report such relationships, I would be more confident in a correlation or KW if you assessed the significance of other possible contributions, and given the limited data presented in this research it could make the manuscript more substantial.
This data was not collected.
Further, wouldn’t a change from normal for individual cows be a more useful/practical indication of lameness ie you might have a cow that is generally low in activity that might not be lame, or you could have a cow that is usually very active suddenly drop in activity, which would be a more practical indication the something is wrong with that animal. This is partly why a clinical assessment of cows identified as lame would have strengthened this study.
Change in behaviour in relation to change in locomotion score was analysed for the 3 trials with 2 locomotion scoring events with similar modest results. Clinical assessment was carried out for trial one (not reported) and initial analysis indicated it was not particularly informative and so was not prioritised for subsequent trials.
Table 1: Please include a footnote where A and B are defined.
This has been implemented.
A lot of the detail in this table (especially the last column) would be better described in the methods.
We believe that while this suggestion has merit, it would be unwieldy text to read and of limited interest to many readers. In the table format, we believe it is clearer.
Table 2. References should be in number format.
Done
Strides, Walking counter, Walking index, Walking time – all examples of variable that would be affected by distance to and from the dairy (which all animals have to do, regardless of mobility), DM offered, weather etc.
Agreed – as would be the case in a commercial application for grazing cows which is likely to make it unpractical. However, the cows in each trial were managed together so we can discern potential differences between cows walking the same distance.
Results
Generally, I think more data needs to presented in the results. At the very least, there is no data on the number of cows in each mobility classification and this must be included.
This has been implemented.
I would also like to see the authors explore the relationships between activity and mobility (1) as a Kruskal wallis non-parametric test, and (2) for the time that cows were at pasture.
As discussed above, we believe Spearmans Rank correlation was appropriate and we have outlined how we performed unreported analysis of cow behaviour between milkings and also found modest results.
External variables (farm, kg pasture DM allocated, weather conditions, time off pasture, distance to the dairy) should also be investigated and eliminated.
We believe some of these variables can’t be controlled for in commercial application. Therefore, accounting for them would significantly reduce commercial/applied relevance of these findings. We have however address weather and parity in this revision which could be implemented into a commercial system.
L103-104: “This was…(distance to paddock)” This is a discussion point and not a result
This has been removed from this sections.
L112-114: “In summary…walking time”. This is also a discussion and not a result
We contend this is simply summarizing the results without discussion.
Table 4: Why are there so many missing data in this Table?
All values between -0.2 and 0.2 were not reported. Another reviewer has suggested printing all the values and so we have done this.
Discussion
L126: delete “in this study” at the end of the sentence
Implemented
L127: are the references provided here for cows at pasture? It would be good to compare/contrast these findings for cows at pasture to similar experiments on cows housed indoors.
A discussion of grazing /non grazing results has been included in LN 72:95.
L129-131: Or, there are things you haven’t controlled for that are washing out the relationships?
This may be case. However, we believe that does not reduce the relevance to commercial applications of lameness detection. Yes a relationship may exist, but if discerning it requires controlling for variables that are not easy to control for in practice, then it is not relevant for a commercial application.
L133: I think the key is to examine changes in behaviour for individual animals, rather than a blanket behavioural assessment.
Change in behaviour analysis was performed (not reported) with similar modest results. We believe the sentence was appropriately limited in its scope. We have now added the following for clarity ‘as they were operationalized in this study’. We note that Beer et al achieve good results but looking at the average of 2 days.
L134-135: sample sizes – these are small sample sizes, and you haven’t provided numbers for the cows within each mobility category you used or on whether the animals studied were randomly chosen or chosen because they were identified as having compromised mobility. This makes it hard to apply your findings to a large herd. For example, on a large herd of cattle at pasture it would be expected that the vast majority of animals would have a mobility score of 0. In differences in behaviour for a cow with a mobility score of 1 or 2 and definitely 3 may be statistically different. But in your studied herds the proportion of animals with impaired mobility and those without I presume (but do not know because the data isn’t presented) would not be representative of a commercial farm. This is why a change in behaviour may be more reliable.
We have addressed this by presenting the numbers of cows with each score in Table 1. We also ran but do not report analysis of change in behaviour for trial 1,3 and 4 and found similar modest results. This analysis and data are available for inspection on github.
L137-138: Again, hard for the reader to assess this as numbers of cows in each category were not reported
This has been addressed. Table 1.
L138-139: Here also, I make the point regarding proportion of animals in each category in the studied group compared to a larger herd and of changes in behaviour perhaps being more appropriate.
This has been addressed. Table 1
L142-146: I think this is an important paragraph and this is why I think you should compare to relationships between behaviour restricted to the times only when cows are at pasture. What do the experiments for cows at pasture report? Are they different to what you have reported? This is also perhaps where a discussion on the usefulness on “changes from normal” behaviour for individual animals.
We found similarly modest results for analysis between milkings and the R script and code are available for inspection.
L149: Ideally you would’ve confirmed the reliability of the observers assessment firstly with a clinical assessment of the animal, and a secondary observer.
Hoof inspections were carried out for the first trial confirming issues with most score 2s and 1s (not reported). Hoof inspections were carried out during the first trial of Jerseys grading each hoof for overgrown, sole haemorrhage and white line disease. These inspections confirmed the presence of issues on each hoof of each score 2 cow with White Line disease being the most prevalent. Each score 1 cow also had issues (usually white line) on each hoof, though they were less severe. Some indications of mild white line disease were also found for most hoofs of cows of score 0. Overall, hoof inspections indicated differences in the prevalence of pathologies somewhat consistent with mobility score.
L150: This is the first time the authors have said that severely lame animals were not included in the study. Was this deliberate? Ideally in a pasture based systems severely lame animals will be moved to sick herd close to the dairy so its usefulness for this cohort of animals is questionable.
The title and abstract identifies the range of severity LN 33, the last paragraph of the introduction now specifies mild and moderate lameness LN96-97. In the herds examined there were almost no score 3 cows, and so we did not included them and focused on mobility issues which are not easy for untrained observers to detect. If we were performing the data analysis again, given the current results, we would consider pulling out 0’s., 1’s and 2’s and putting them in with the sick herd with score 3 cows so that severe lameness could be assessed also.
L153: And what did Blackie find? The same as the present research? Did Blackie study animals at pasture?
They found mixed results with housed animals. ‘Blackie et al [22] reported differences in lying time between severely and moderately lame grouped together and non-lame zero grazing Holstein cows and Blackie and Maclaurin [23] later found that similar associations were not present in a study of zero grazing Jersey cows. Behavior lameness associations could thus fail to generalize depending on the severity of lameness assessed, if the cows are zero grazing/grazing and the breeds studied.’ LN 88
L161-163: This sentence is repetitive of early paragraphs
The discussion section has been significantly revised.
L165-166: I would like to see the authors expand on this discussion point, as it is important. Have no behaviours been found useful in the non-grazing animal? Or some behaviours? How many studies have found no association? Have any reported contrary results? If there truly are no relationships for mild/moderate lameness in either system, then why? Is the animals behaviour just not affected enough by the condition at this early stage?
The introduction Ln 72:95 and the new limitations section LN 227:261 now discusses grazing/non-grazing and severity in more detail.
L168-169: I think you can delete the first sentence of this paragraph.
This has been implemented.

Reviewer 3 Report
This study evaluates whether automated lameness detection system is feasible to find mildly or moderately impaired mobility of grazing cows. Overall, the subject of this study sounds reasonable and interesting. I agree with the idea that automatic lameness detection systems should be tested under a wide range of conditions. The findings of this study will provide notice to researchers and engineers when developing a lameness detection system. Thus, the manuscript has a merit to publish in this journal. However, the manuscript includes several concerns:
1. Although the study targets "grazing cows", the authors did not show any problem with grazing when detecting mildly and moderately impaired mobility of cows. The authors should refer to it in the introduction section
2. Pasture condition and grazing management in each trial should be summarized in a table and running text (line 78–80).
3. This study adopted a 0 to 3 scale of lameness score and excluded score 3 (severely lame cows) from the analysis. If this is true, you use 3-point-scale to analyze the relationship between cow’s behavior and lameness. My concern is that such a rough scale is the reason not to detect any relationship between the behavior and lameness. In addition, the result of the lameness score in each trial did not show in the current manuscript. These data should be reported in the revised version.
4. Generally, if you do not detect a significance in correlation analysis, it means that you do not reject the null hypothesis, i.e. r = 0. Thus, even if "r coefficient" itself seems to show some relation between variables, it does not mean anything. I believe that this fact does not affect your findings and conclusion, but some of results and discussion should be revised.
5. According to Table 1, timing of scoring and behavior recording varies with the trials. It might be inevitable when collecting data under actual conditions, but the authors should add some explanations.
Details
Lines 113–114: "Walking counter" also shows non-significant association (-0.2) with mobility score in average (but see the next comment).
Line 124–131: As the authors suggested in the introduction section, some studies revealed that walking duration, lying bout and other behavior variables are associated with lameness. So, the inconsistency between the previous studies and the present study should be discussed.
Lines 128–131: As mentioned earlier, a non-significant correlation does not support any relationship between variables in a conservative view. Thus, you should reconsider this statement. In contrast, some cases showed clear relationship, e.g., limb events in 4a, standing index in 4a and walking index in 2. The authors should refer to these variables, even if you consider these results do not have any meanings.
Line163–166: The authors should refer to the issue why detecting lameness automatically is challenging in a grazing condition.
Table 4: In my opinion, all values including low coefficients (-0.2 to 0.2) should be reported in this Table. Scatter plots of these correlation should be presented in a supporting figure. This is important to consider the validity of this analysis.
Author Response
Please find attached a revised version of the manuscript. Below a letter to the reviewers. Below that, responses to specific reviewer queries.
Letter to the reviewers following first round of comments
Dear Reviewers,
We would like to thank you for your thoughtful, thorough, robust and very constructive input. We have revised the paper significantly in light of these comments including the reporting of additional data, power analysis and analysis of aggregated data. We have rewritten most of the paper and so do not mark changes as this would be mark than 70% of the paper. In particular, we discuss in more detail the role of grazing / zero grazing, severe/ non-severe lameness and breed. Upon further reflection, we also clarify that the papers’ value is not in discerning if associations exist in controlled conditions – but rather asking
‘are the studied behaviors likely to be useful for automated lameness detection in the studied commercially relevant contexts?’
Though the main findings of the initial submission remain broadly similar there are some changes. In performing the additional analysis requested by reviewer 1, 2 coding errors were identified and are now rectified. The first of these was an error in trial one matching cows to recorded behaviour, and the other in later trials which excluded some cows incorrectly. Some other cows are now excluded as a closer inspection of the data identified unfeasibly low walking time data. As such the sample sizes of the individual trials reported has changed (16 +16+ 16+15 = 63). With additional aggregated data analysis we now report a significant association between standing up/laying counter activity and lameness.
We have also clarified the strengths and limitations of the reported studies. We wish to highlight here the strengths - multiple trials, multiple breeds, multiple farms, open access data and statistical analysis code. In the discussion, a limitation section now includes a discussion of parity and acknowledges this study would ideally have had greater statistical power. Reviewer 2 recommended accounting for / controlling for a range of variables including parity, distance to paddock and weather. Of these, parity has been identified as a confounder and it could be integrated into a commercial system relatively easily. However, we did not find any associations with behaviour and lameness in our study. Regarding weather, precipitation is the main concern in the literature and this was minimal for the studied days and so did not affect this study.
For the other variables e.g. distance to paddock or feed allowance - it would be a challenge to include these in a commercially viable automated system. We would thus see these variables as necessarily being non-essential/optional for a commercially viable lameness detection system. If a behaviour is likely to be useful for automated detection in pasture based systems, there should be statistically robust correlations without having to control for several variables. Controlling for variables should be seen as a way to increase performance – not essential to discerning any statistically robust difference.
With the possible exception of standing up/laying counter and activity, we still believe the results indicate the studied variables are likely to be of limited use for lameness detection in the studied context. Though not reported in the paper, we also analysed the following and found similar results to that with the basic analysis.
- behaviour between milkings,
- ratio of day to night time behaviour,
- change in behaviour and locomotion for 3 trials, and,
- measures of variance.
We have published the code and data required to run this analysis on our github for reviewers interested in running this analysis themselves and viewing the results. https://github.com/nialloleary/Behavior-Lameness-Article
Based on our initial reading of the positive sounding literature, we were disappointed we did not find positive results. This studies findings are, however, congruent with this review some of the authors recently contributed to. (https://www.sciencedirect.com/science/article/pii/S0022030220301417?via%3Dihub). Progress by focusing on behaviour has been limited and we believe this is in large part because of a failure to replicate and failure to acknowledge those findings reported are relatively modest in effect size and inconsistent. In the review, we conclude behaviour should no longer be the main focus. We further argue that analysis of gait should be the focus of further research (e.g. Halahchen et al, Alsaaod et al) and a combination of gait and behaviour, with a focus on gait is the most promising avenue for accelerometer based lameness detection.
All 3 reviewers engaged constructively and thoughtfully with this paper. Thank you again for your excellent contributions and we look forward to any further remarks you may have. We hope we have address most, if not all those raised so far to your satisfaction.
Regards,
Niall O’Leary
Reviewer 3
This study evaluates whether automated lameness detection system is feasible to find mildly or moderately impaired mobility of grazing cows. Overall, the subject of this study sounds reasonable and interesting. I agree with the idea that automatic lameness detection systems should be tested under a wide range of conditions. The findings of this study will provide notice to researchers and engineers when developing a lameness detection system. Thus, the manuscript has a merit to publish in this journal. However, the manuscript includes several concerns:
- Although the study targets "grazing cows", the authors did not show any problem with grazing when detecting mildly and moderately impaired mobility of cows. The authors should refer to it in the introduction section
Is the reviewer referring to lameness prevalence in grazing systems? Or the difficulty of scoring cows at pasture?
- Pasture condition and grazing management in each trial should be summarized in a table and running text (line 78–80).
This data was not aggregated, as it would be hard to incorporate into a commercially viable product.
- This study adopted a 0 to 3 scale of lameness score and excluded score 3 (severely lame cows) from the analysis. If this is true, you use 3-point-scale to analyze the relationship between cow’s behavior and lameness. My concern is that such a rough scale is the reason not to detect any relationship between the behavior and lameness.
This is a fair point highlighting the inadequacy of using locomotion score as a reference. We considered hoof inspections and did this for trial 1 (not reported) but this is also complicated to operationalize. Score 2 cows should however be discernible from score 0 cows in any automated system worth using.
In addition, the result of the lameness score in each trial did not show in the current manuscript. These data should be reported in the revised version.
The herd prevalence was not calculated. The studied sample prevalence is now included in Table 1.
- Generally, if you do not detect a significance in correlation analysis, it means that you do not reject the null hypothesis, i.e. r = 0. Thus, even if "r coefficient" itself seems to show some relation between variables, it does not mean anything. I believe that this fact does not affect your findings and conclusion, but some of results and discussion should be revised.
The discussion has been revised including accounting for the statistical power of the study.
- According to Table 1, timing of scoring and behavior recording varies with the trials. It might be inevitable when collecting data under actual conditions, but the authors should add some explanations.
This has been included ‘For trials 1, 3 & 4, cows were mobility scored at the beginning and the end of these trials and each scoring event is referred to as ‘A’ and ‘B’ respectively (Table 1). The time between scoring events ‘A’ and ‘B’ varied with operational requirements from 4 days to 14 days. In trial 2, cows were only scored once due to locomotion scorer unavailability.’ LN128:132
Details
Lines 113–114: "Walking counter" also shows non-significant association (-0.2) with mobility score in average (but see the next comment).
Line 124–131: As the authors suggested in the introduction section, some studies revealed that walking duration, lying bout and other behavior variables are associated with lameness. So, the inconsistency between the previous studies and the present study should be discussed.
The limitations and implications sections of the discussion now discusses these points.
Lines 128–131: As mentioned earlier, a non-significant correlation does not support any relationship between variables in a conservative view. Thus, you should reconsider this statement. In contrast, some cases showed clear relationship, e.g., limb events in 4a, standing index in 4a and walking index in 2. The authors should refer to these variables, even if you consider these results do not have any meanings.
We hope the discussion of statistical power in the paper now addresses this concern LN164:173.
Line163–166: The authors should refer to the issue why detecting lameness automatically is challenging in a grazing condition.
The implications section of the discussion addresses this now.
Table 4: In my opinion, all values including low coefficients (-0.2 to 0.2) should be reported in this Table.
Implemented
Scatter plots of these correlation should be presented in a supporting figure. This is important to consider the validity of this analysis.
Figure 1 is now a boxplot similar to this suggestion.
